# OpenMedQ: Broad Open Pretraining for Medical Vision-Language Models

**Ibrahim Gulluk**[*1]                                    GULLUK@STANFORD.EDU

**Max Van Puyvelde**[*2,3]                          MAXVPUYV@STANFORD.EDU

**Olivier Gevaert**[2]                                    OGEVAERT@STANFORD.EDU

[1] *Department of Electrical Engineering, Stanford University*

[2] *Department of Biomedical Data Science, Stanford University School of Medicine*

[3] *Department of Mathematical Modelling, Statistics and Bioinformatics, Ghent University*

## Abstract

We present *OpenMedQ*, a medical vision-language model pretrained on the broadest fully-open medical mix to date: 14 datasets totaling ∼3.35M pretraining samples spanning pathology, radiology, microscopy, and text-only clinical QA. OpenMedQ reaches state-of-the-art BLEU-1 on PathVQA (75.9), beating Med-PaLM M variants up to 562B parameters (∼80× larger), and matches the best reported VQA-MED BLEU-1 (64.5). Its vision encoder, transferred to 8 unseen medical classification benchmarks under an identical downstream recipe, obtains the highest average macro-F1 (0.757) among BiomedCLIP (0.745), PMC-CLIP (0.745), PubMedCLIP (0.746), and a from-scratch baseline (0.616). We release our code and an interactive demo is publicly available as a reproducible baseline for the community.

**Keywords:** Medical Vision-Language Models, Medical Image Classification, Open Science

## 1. Introduction

Medical foundation models are increasingly capable, yet most published medical VLMs rely on a handful of narrow pretraining sources and withhold either their weights, their data, or both. Contrastive encoders such as BiomedCLIP (Zhang et al., 2023b), PMC-CLIP (Lin et al., 2023), and PubMedCLIP train on single image-caption corpora; generative medical VLMs such as PMC-VQA (Zhang et al., 2023c) and LLaVA-Med (Li et al., 2024) demonstrate strong visual question answering (VQA) on a few benchmarks but use comparably narrow pretraining mixes, while BiomedGPT (Zhang et al., 2023a) and Med-PaLM M (Tu et al., 2024) scale data and parameters but do not release weights. This leaves practitioners without a fully-open, broadly-pretrained baseline they can actually inspect, reuse, and extend.

We introduce *OpenMedQ*, a LLaVA-style (Liu et al., 2024) VLM (ViT-base (Zhang et al., 2023b) + LLaMA-7B (Touvron et al., 2023; Wu et al., 2024), LoRA (Hu et al., 2021)) trained on the broadest open medical pretraining mix to date (14 datasets, ∼3.35M samples) with next-token prediction. We will release weights and dataset recipes upon acceptance; a live interactive demo is already available at https://openmedq.streamlit.app/ for qualitative inspection.

---

[*] Equal contribution

## 2. Method

**Architecture and pretraining.**  The vision encoder $f_{\text{vis}}$ is a ViT-base-patch16-224 initialized from BiomedCLIP (Zhang et al., 2023b); a linear projection feeds its image tokens into a LLaMA-7B (Touvron et al., 2023) language model initialized from PMC-LLaMA (Wu et al., 2024). Image and text tokens are concatenated and decoded left-to-right, following LLaVA (Liu et al., 2024). We fine-tune with LoRA (Hu et al., 2021) of rank $r = 8$ using next-token cross-entropy with image and prefix tokens masked. All images are resized to 224×224; training uses AdamW, batch size 64, learning rate $5{\times}10^{-5}$, for up to 15 epochs on a single NVIDIA A100.

**Classification transfer.**  To probe the vision features produced by pretraining, we detach $f_{\text{vis}}$ and attach a linear head $W \in \mathbb{R}^{2d \times m}$; encoder and head are fine-tuned together on each downstream dataset for 100 epochs. We benchmark OpenMedQ's encoder against three strong medical contrastive baselines (BiomedCLIP, PMC-CLIP, PubMedCLIP) and a from-scratch baseline, all under an identical downstream recipe so that any gap is attributable to the pretraining.

## 3. Datasets

**Pretraining mix (14 datasets, ∼3.35M samples).**  Image-text sources (∼2.94M pairs) span pathology (PathVQA (He et al., 2020)), radiology (VQA-RAD (Lau et al., 2018), IU-XRAY (Demner-Fushman et al., 2016), MIMIC-CXR (Johnson et al., 2019), ROCO (Pelka et al., 2018), OmniMedVQA (Hu et al., 2024)), mixed modalities (Slake (Liu et al., 2021), PMC-OA (Lin et al., 2023), PMC-VQA (Zhang et al., 2023c), VQA-MED (Ben Abacha et al., 2019)), and microscopy ($\mu$-Bench (Lozano et al., 2024)). A further ∼410K text-only clinical QA samples (MedQA, MedMCQA, PubMedQA) are included to preserve language capability during pretraining.

**Classification benchmarks (8 datasets).**  We evaluate on CXR8 (Wang et al., 2017), MedFMC (med, 2023) (chest, colon, endo subtasks), Breast-Ultrasound (Al-Dhabyani et al., 2020), CHAOYANG (Zhu et al., 2021), CBIS-DDSM (Lee et al., 2017), and Mendeley-CXray (Kermany et al., 2018). These datasets were not seen during pretraining.

## 4. Results

**Classification transfer.**  Figure 1(a) is our headline result. OpenMedQ achieves the highest mean macro-F1 (0.757) across the eight benchmarks, ahead of PubMedCLIP (0.746), PMC-CLIP and BiomedCLIP (0.745), and the from-scratch baseline (0.616). OpenMedQ wins outright on MedFMC-chest and MedFMC-endo, ties PMC-CLIP on CXR8, and trails the best encoder by at most 0.02 on four more; the only meaningful gap is Breast-Ultrasound (0.876 vs. 0.915). Since the downstream recipe is fixed, this delta reflects what OpenMedQ's pretraining added to the BiomedCLIP initialization.

**Open-ended VQA.**  On PathVQA, OpenMedQ reaches 75.9 BLEU-1, beating prefix tuning (Van Sonsbeek et al., 2023) (70.3) and all three Med-PaLM M variants up to 562B (Tu

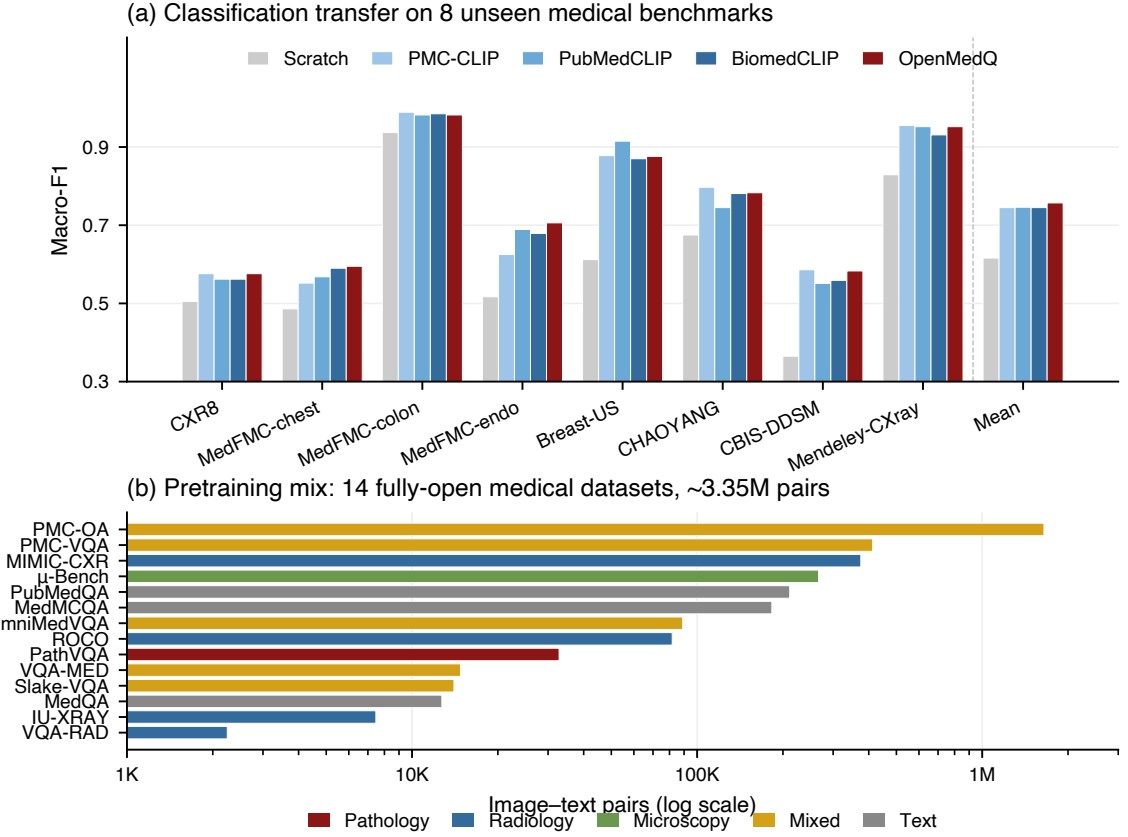

Figure 1: **(a)** Macro-F1 across 8 unseen medical classification benchmarks: all bars share an identical downstream recipe and differ only in the pretrained vision encoder. OpenMedQ attains the highest *Mean* (0.757). **(b)** OpenMedQ's pretraining mix: 14 fully-open datasets (∼3.35M pairs), colored by modality group.

et al., 2024) (72.27) despite using only 7B parameters. On VQA-MED, OpenMedQ reaches 64.5, just above the 2019 challenge best (64.4).

## 5. Discussion

*Breadth* of open pretraining data is a competitive lever for medical VLMs: at 7B parameters, OpenMedQ sets a new state of the art on PathVQA against Med-PaLM M up to 562B, and its vision encoder beats three strong contrastive medical encoders on average classification transfer. Data diversity is a reproducible lever; proprietary scale is not. The lever has its limits: Med-PaLM M's larger variants still lead on VQA-RAD and Slake, BLEU-1 captures only surface agreement, and narrow-modality encoders can edge us out on Breast-Ultrasound. The demo is available at https://openmedq.streamlit.app/.

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
