# OpenReview forum: "OpenMedQ: Broad Open Pretraining for Medical Vision-Language Models"
_MIDL.io/2026/Short_Papers — MIDL 2026 - Short Papers Poster_

### Official Review · Reviewer_EZ1J · 2026-04-23
**Review for OpenMedQ: Broad Open Pretraining for Medical Vision-Language Models**

**Rating:** 5
**Confidence:** 4

**Review:**

This paper presents a nice study on VQA with a traceable training dataset and a large set of downstream tasks and datasets. It clearly describes the training procedure and architectural setup. It also includes an online demo to showcase the model's generated output. Its comparison and evaluation are comprehensive. Methodologically, the paper could be extended with more ablations and explanations of design choices.

**Summary:**

The authors present OpenMedQ, a medical VQA with open source weights and focus on reproducibility in terms of data traceability.
Methodologically it builds upon open source foundation models such as LLama and BiomedCLIP. It outperforms other VQA method on unseen data classification on eight datasets. In open ended VQA it outperforms MeD-PaLM M variants.

**Strengths:**

- the authors claim that they will open source their model which should foster further research and development
- There is a publically available web demo that shows precomputed outputs of the model
- The model was trained and evaluated on a diverse set of datasets coming from different modelities.
- The model outperforms a substantial number of well known VQA models

**Weaknesses:**

- Using the model in their demo shows slight signs of overfitting. From a lateral view of the chest, the model responses with pa and lateral view. Might be related to a bias in the dataset as well that many images are taken together, but simply from that image the model should not talk about the lateral view
- The choice of models for comparison used to benchmark seems selective. For example Fig 1b) shows no results on Med-PaLM M. What about MedGemma which is not mentioned at all
- There is no ablation or justification of the design choices of the training. Providing more details could help future researchers learn important design aspects of VQA models

**Justification Of Rating:**

This paper proposes a VQA model that reaches performance comparable to other large VQA models that are behind paywalls, closed-source, or trained with unclear data. It fosters the development of locally trainable and deployable VQA models and will be the basis of great discussions at the conference. I recommend acceptance.

---

### Decision · Program_Chairs · 2026-05-08

Accept (Poster)